# Non-Invasive and Quantitative Estimation of Left Atrial Fibrosis Based on P Waves of the 12-Lead ECG—A Large-Scale Computational Study Covering Anatomical Variability

**DOI:** 10.3390/jcm10081797

**Published:** 2021-04-20

**Authors:** Claudia Nagel, Giorgio Luongo, Luca Azzolin, Steffen Schuler, Olaf Dössel, Axel Loewe

**Affiliations:** Institute of Biomedical Engineering, Karlsruhe Institute of Technology (KIT), Kaiserstr. 12, 76131 Karlsruhe, Germany; giorgio.luongo@kit.edu (G.L.); luca.azzolin@kit.edu (L.A.); steffen.schuler@kit.edu (S.S.); Olaf.Doessel@kit.edu (O.D.); axel.loewe@kit.edu (A.L.)

**Keywords:** electrophysiological simulation, atrial cohort modeling, 12-lead ECG, P wave, atrial fibrosis, atrial fibrillation

## Abstract

The arrhythmogenesis of atrial fibrillation is associated with the presence of fibrotic atrial tissue. Not only fibrosis but also physiological anatomical variability of the atria and the thorax reflect in altered morphology of the P wave in the 12-lead electrocardiogram (ECG). Distinguishing between the effects on the P wave induced by local atrial substrate changes and those caused by healthy anatomical variations is important to gauge the potential of the 12-lead ECG as a non-invasive and cost-effective tool for the early detection of fibrotic atrial cardiomyopathy to stratify atrial fibrillation propensity. In this work, we realized 54,000 combinations of different atria and thorax geometries from statistical shape models capturing anatomical variability in the general population. For each atrial model, 10 different volume fractions (0–45%) were defined as fibrotic. Electrophysiological simulations in sinus rhythm were conducted for each model combination and the respective 12-lead ECGs were computed. P wave features (duration, amplitude, dispersion, terminal force in V1) were extracted and compared between the healthy and the diseased model cohorts. All investigated feature values systematically in- or decreased with the left atrial volume fraction covered by fibrotic tissue, however value ranges overlapped between the healthy and the diseased cohort. Using all extracted P wave features as input values, the amount of the fibrotic left atrial volume fraction was estimated by a neural network with an absolute root mean square error of 8.78%. Our simulation results suggest that although all investigated P wave features highly vary for different anatomical properties, the combination of these features can contribute to non-invasively estimate the volume fraction of atrial fibrosis using ECG-based machine learning approaches.

## 1. Introduction

Atrial Fibrillation (AF) is the most common cardiac arrhythmia worldwide and affects 2–3% of the European and North American population [1]. The clinical picture of AF is characterized by disorganized re-entrant waves traversing the atrial tissue and causing an irregular heart rhythm. The initiation and maintenance of the abnormal electrical activity requires ectopic foci firing and regions of vulnerable atrial substrate. Hence, fibrotic tissue having undergone structural and electrical remodeling processes provides the necessary substrate properties to contribute to the perpetuation of AF [2,3]. Quantifying the amount of these arrhythmogenic substrate areas could thus be an important means for individual risk stratification of new-onset AF and arrhythmogenic fibrotic atrial cardiomyopathy (FAM) [4,5].

Catheter ablation is a common treatment option in clinical practice for the purpose of blocking certain conduction pathways in the atria that are suspected to contribute to the onset and maintenance of the arrhythmia. However, the optimal choice of these ablation targets beyond pulmonary vein isolation (PVI) is challenging and calls for a personalized approach. AF recurrence rates between 20 and 60% in patients with large arrhythmogenic substrate areas in the atria [6] underline the need to tailor therapy to the substrate.

High intensity areas on late Gadolinium enhancement magnetic resonance images (LGE-MRI) as well as low bi- and unipolar electrogram voltages [7,8,9] during electrophysiological studies are currently drawn on to identify the amount and spatial location of scars and fibrosis on the atrial wall. However, LGE-MRI is a cost-intensive imaging technique for which technical parameters have to be carefully selected and the segmentation of MR images is cumbersome and challenging [10]. Furthermore, electroanatomical mapping is an invasive and time-consuming procedure with inter-electrode spacing and electrode sizes being additional confounding factors influencing electrogram voltages [11].

To circumvent these shortcomings, the 12-lead electrocardiogram (ECG) as a commonly available and easily usable tool in clinical practice could feature a way to quantify the fibrotic left atrial (LA) volume fraction and in consequence predict the propensity to prospective AF incidents. The duration of the P wave in the 12-lead electrocardiogram (ECG) has been shown to correlate with the amount of LA low-voltage substrate in AF patients if a low threshold for determining P wave on- and offsets are chosen [5,12,13]. P wave terminal force in V1 (PTF V1) has been identified as sensitive to a change in conduction velocity in the atria [14], which is typically caused by fibrotic substrate in AF patients. Furthermore, P wave dispersion has been shown to be a measure for locally heterogeneous conducting tissue regions [4]. Other ECG-based predictors for the onset of AF and for the presence of low-voltage areas include P wave amplitude [15,16] and P wave area in V3 [17].

However, these P wave derived features are not only affected by fibrotic infiltration of atrial tissue. P wave duration (PWD) and dispersion are also influenced by changes in conduction velocity and atrial anatomy [18] that both vary between different subjects within healthy reference ranges. Furthermore, PTF V1 is highly dependent on the placement of the electrodes on the thorax [19]. Lastly, different thoracic geometries yield different P wave amplitudes, which might additionally be impaired by loose electrode contact.

In this study, we therefore aim to quantify which changes in P wave morphology, amplitude and duration can be attributed specifically to fibrotic infiltration of atrial tissue. Furthermore, we investigate if these effects on the P wave can be separated from confounding changes induced by healthy anatomical variation of the atria and the thorax as well as different electrode positions. For that purpose, we conducted electrophysiological simulations on varying anatomical models in sinus rhythm with and without inclusion of fibrotic regions in the atria of different volume fraction. Subsequently, changes in P wave features caused by geometry and rotation angle variations were compared to those caused by local atrial substrate modifications. The results thereof are employed to estimate the fibrotic LA volume fraction from P wave features.

## 2. Materials and Methods

### 2.1. Database

In order to generate a large-scale dataset of P waves representing a virtual patient cohort characterized by different anatomical properties, we derived various atrial and thoracic geometries from statistical shape models (SSM). We used the bi-atrial SSM previously developed in our group [18] to generate 80 random volumetric instances of the atria augmented with homogeneous wall thickness, rule-based fiber orientation [20], tags for anatomical structures and inter-atrial connections [21]. One exemplary instance is shown in Figure 1.

Furthermore, 25 thoracic geometries were generated by varying the two leading eigenvectors of the model developed by Pishchulin et al. [22] systematically in the [−2,2]σ range. The first two eigenvectors account for approximately 80% of the total shape variance. The first two eigenmodes, i.e., the shape variability resulting from a variation of solely the respective eigenvector, are visualized in Figure 2. A variation of both of them reflects in a change of the torso size in superior-inferior direction and in anterior-posterior direction in both the chest and the abdominal regions.

Each of the 80 atrial geometries were rotated by −20${}^{\circ }$, 0${}^{\circ }$ and +20${}^{\circ }$ around the x-, y- and z-axes [23] leading to 27 permutations of different orientation angles for each individual atrial geometry. Combinations of every rotated atrial geometry placed in each of the 25 thoracic geometries were realized and thus a set of 54,000 anatomical models was obtained. The different model and parameter combinations are illustrated in Figure 3. For each of these combinations, electrophysiological simulations were conducted in sinus rhythm (SR) and the 12-lead ECG was extracted at the standardized electrode positions. The SR simulations were repeated with electrical and structural remodeling of different degrees for all model combinations. For that purpose, the ionic and tissue parameters were modified as described in Section 2.2 in selected tissue patches covering 0%, 5%, 10%, 15%, 20%, 25%, 30%, 35%, 40% and 45% of the total LA myocardial volume. In this way, we obtained a total of 540,000 12-lead ECG signals (80 atria geometries × 25 torso geometries × 27 rotation angles × 10 volume fractions covered by fibrosis) subject to the analysis described in Section 2.4 and Section 2.5.

### 2.2. Modeling Methodology of Fibrotic Tissue

For each atrial model, we created variants with fibrosis covering 0%, 5%, 10%, 15%, 20%, 25%, 30%, 35%, 40% and 45% of the total LA myocardium volume. Subsequently, the volume fraction of right atrial fibrosis was defined for each case according to the findings by Akoum et al. [24] (Table 1).

Considering the patchiness of fibrosis observed in AF patients [25], we defined several disconnected patches on the atrial surface as fibrotic accumulating to the total LA volume fraction of fibrosis. Each of these individual fibrotic patches is defined by a center seed point and a radius around it. The total number of seed points and the sizes of the radii were chosen depending on the total volume fraction of fibrosis to be covered. The positions of the seed points for the patchy fibrotic regions were defined by taking into account the spatial distribution of fibrotic atrial substrate as reported by Higuchi et al. [26] for the left and Akoum et al. [24] for the right atrium. Radii randomly chosen in a range of [3, 6] mm around these seed points determined the candidate regions of fibrosis in the atria. To not only account for the patchy nature of fibrotic tissue, but also for its diffuse appearance, we assigned ≈80% of the cells within the candidate regions defined above as belonging to the fibrotic substrate (compare Figure 4 (middle and left column)). In these substrate regions, the simulation parameters were adjusted as described in the following to account for structural and electrical fibrotic remodeling.

Fibrosis infiltrating the regular myocyte tissue structure cause adjacent myocytes to be electrically decoupled and thus act as passive barriers to the propagating wavefront. We draw on the concept of percolation [27] to account for this phenomenon in our simulations. In total, 50% of the cells within the fibrotic regions were therefore defined as belonging to the extracellular matrix. Hence, these cells impair the normally straight conduction along the tissue and constrain the intracellular depolarization wave to pass around the fibrotic barriers [28]. In the remaining 50% of the cells belonging to the fibrotic regions, maximal ionic conductances were rescaled as suggested by Roney et al. [29] to account for cytokine-induced remodeling (50% g${}_{K_{1}}$, 60% g${}_{Na}$, 50% g${}_{CaL}$) [30]. Furthermore, conduction velocities were reduced by 80% in transversal and 50% in longitudinal direction, which in turn caused anisotropy ratios to be increased by a factor of 2.5. In this way, we accounted for local conduction velocity (CV) heterogeneities and anisotropic wavefront propagation facilitating functional re-entry in AF patients [31].

### 2.3. Electrophysiological Simulations

For each atrial model and volume fraction covered by fibrosis, simulations were performed by solving the anisotropic Eikonal equation with the fast iterative method (https://github.com/KIT-IBT/FIM_Eikonal, accessed on 21 April 2021) [32]. For all simulations, excitation was initiated from a sinus node trigger located at the junction of the superior caval vein and the right atrial appendage [33]. The atrial wall was separated into five different anatomical regions: bulk right and left atrium, inter-atrial connections, pectinate muscles, crista terminalis and inferior isthmus. For each of these regions, the anisotropy ratio and conduction velocity in transversal fiber direction CV${}_{⊥}$ were chosen as reported previously [33] (Table 2). For simulations involving fibrotic tissue areas, two additional anatomical regions were included as described in Section 2.2: Non-conductive elements were characterized by a conduction velocity of CV${}_{⊥}$ = 0 mm/s and CV was reduced by 80% in slow conducting cells. Anisotropy ratio in slow conducting tissue was increased by a factor of 2.5 compared to baseline.

By solving the Eikonal equation, we computed the spread of electrical activation in SR and obtained local electrical activation times (LATs) at each node. By shifting a Courtemanche et al. [34] action potential template in time according to the calculated LATs as proposed by Kahlmann et al. [35], the transmembrane voltage distribution on the atria was obtained. A remodeled Courtemanche action potential as described in Section 2.2 was used for cells in slow conducting fibrotic areas representing cytokine-induced remodeling.

For each model combination explained in Section 2.1, we assumed the atria to be embedded in an infinite volume conductor of conductivity $\sigma_{s}$ = 0.2 S/m. The extracellular potentials were extracted at the respective electrode positions [36] and used to derive the P wave of the standard 12-lead ECG. For analyzing the influence of the V1 electrode position on PTF V1, we also extracted the 12-lead ECG from the electrophysiological simulations with the reference atria and torso geometry for varying positions of the V1 electrode. The latter was varied within a radius of 6 cm around the standard V1 electrode position.

### 2.4. ECG Analysis and Feature Extraction

For each of the resulting 540,000 12-lead P wave ECGs, the following features were calculated: duration, dispersion, terminal force in V1 and peak to peak amplitude in each lead. These features have been shown to correlate with the presence of fibrotic atrial tissue in previous studies [5,14,15,16,17].

P wave duration (PWD) was calculated as the time difference between the latest detectable P wave ending and the earliest detectable P wave beginning across all 12 leads. For that purpose, we defined the P wave beginning and ending for each channel individually. In contrast to clinically recorded signals, solely the actual electrical activity originating from the depolarization of the atria reflects in the simulated P waves and no interfering noise sources superimpose the signal. Therefore, the P wave endings and beginnings were annotated with simple thresholds defined above the isolelectric line. P wave dispersion was subsequently derived by computing the time difference between the latest and earliest detected P wave ending across all 12 leads. To calculate PTF V1, the time difference between the detected P wave ending in V1 and the signal crossing the isolelectric line between positive and negative deflection was multiplied with the minimum amplitude in V1. The peak-to-peak amplitudes were obtained by subtracting the minimum from the maximum P wave signal value in all 12 leads individually.

### 2.5. Regression Using Neural Networks

To evaluate whether and to what extent the influence of healthy inter-individual anatomical variability on the P wave can be separated from changes caused by atrial fibrosis, a regression neural network was set up. We used the function fitting neural network in MATLAB (Version 2020b, MathWorks Inc., Natick, MA, USA) with one hidden layer comprising 10 neurons. Bayesian regularization served as a training algorithm to predict the volume fraction of LA fibrosis. In total, 15 P wave derived features described in Section 2.4 (duration, dispersion, PTF V1, peak to peak amplitude in all 12 leads) served as input for the network. To evaluate the network’s performance, we used 70% of the data for training, 15% for validation and 15% for testing. P waves generated with one specific atrial geometry were never assigned to different sets but kept all in the same split (‘pseudo-random split’) as proposed by Luongo et al. [37]. This procedure ensured that the network is blinded for previously unseen atrial geometries during testing and does not generate look-up tables to link P wave features in the test set to nearly the same feature values in the training set caused, e.g., only by a slight rotation of the atria. Luongo et al. [37] also found that excluding thoracic instead of atrial geometries during training still leads to good generalization results, which is why we decided for the pseudo-random split using certain atrial geometries exclusively during testing. The network was trained 10 times with pseudo-random train-validation-test splits for 1000 epochs each. Its performance was assessed by taking the mean of the absolute root mean square error (RMSE) between the predicted and the actual volume fraction of LA fibrosis among all 10 training iterations.

To calculate receiver operating characteristics (ROC) and provide sensitivity and specificity metrics, we set up two scenarios for assigning the model cohorts to different classes. For the first one, we used the fibrotic LA volume fraction limits summarized in Table 1 to assign the respective Utah class to each sample [38]. For the second one, we simply subdivided the virtual cohorts into a healthy control group (0–5% fibrotic LA volume) and a group with fibrotic atrial cardiomyopathy (≥5% fibrotic LA volume).

## 3. Results

The resulting P wave features were analyzed regarding three different aspects: First, the influence of anatomical variabilities and electrode positions on the P wave features were compared to those caused by the presence of atrial fibrosis (Section 3.1). Afterwards, we investigated to what extent the volume fraction of fibrotic substrate resulted in altered P wave features (Section 3.2). In Section 3.3, we analyzed if the effect of healthy anatomical variations on P wave features can be separated by a neural network from the feature changes resulting from the presence of fibrosis.

### 3.1. Influence of Geometries, Rotation Angles and Electrode Positions on P wave Features

The individual influence of anatomical variations and the placement of electrodes on P wave derived features was assessed by varying one of these factors at a time while keeping the remaining ones constant at their reference value. In this way, the variance of each P wave feature resulting from a change of the atrial geometry, the torso geometry, the atrial rotation angle and in the case of PTF V1 also the position of the V1 electrode was analyzed.

Figure 5 shows the P wave feature distributions. The color code represents the dominant change underlying a variation of the respective factor, e.g., LA volume as a key property of atrial geometry alterations. The other dominant properties were the torso size in anterior-posterior direction, the rotation angle around the z-axis and the position of the V1 electrode in inferior-superior direction for a variation of the thoracic geometry, the rotation angle and the V1 electrode placement, respectively. To gauge the potential of one specific P wave feature to be a predictor for the fibrotic atrial volume fraction, also the P wave feature values resulting from fibrotic infiltration in the reference geometry are shown aside with the color code representing the volume fraction of fibrosis covering the total LA tissue volume.

Figure 5a reveals that all examined factors have an impact on the value of PTF V1. The largest variance is however caused by a change in the torso geometry. The vertical line in Figure 5a represents the threshold value of 4 mV·ms typically used when diagnosing structural heart abnormalities. Thoracic variations characterized by a low diameter in AP direction and atrial variations holding large LA volume values yielded increased PTF V1 values, with some of them even above the threshold of 4 mV·ms. The atrial rotation angle and a rather superior placement of the V1 electrode on the thorax also caused higher PTF V1 values. The interquartile ranges in Figure 5a shows that the fibrotic LA volume fraction had a smaller effect on this feature than any anatomical variation. PWD was not affected by torso size and rotation angle (Figure 5b). On the other hand, large LA volumes yielded P waves with durations up to 130 ms (Figure 5b). Electrical and structural remodeling of fibrotic tissue resulted in PWDs up to 160 ms. Due to its proximity to the LA lateral wall characterized by a high probability for the presence of fibrosis, the peak-to-peak amplitude is exemplary shown for lead V6 in Figure 5c. All four analyzed factors affected this feature, while the torso geometry caused the largest variation. P wave dispersion was mostly affected by the atrial geometry (Figure 5d). The maximum dispersion was 28 ms for the healthy anatomical variations and 27 ms in the presence of fibrosis (Figure 5d).

### 3.2. Effect of the Fibrotic LA Volume Fraction on P Wave Features

To examine to which extent the *specific volume fraction* of local atrial substrate modification causes graded changes in the P wave features, each set of 54,000 12-lead ECGs belonging to one particular volume fraction of fibrosis was analyzed at a time. The resulting feature values in dependence of the LA fibrosis volume fraction are shown in Figure 6. Increasing volume fractions of LA fibrosis systematically caused changes of the feature values. However, when considering all anatomical variations as it is the case in Figure 6, the value ranges of all features overlap between the healthy and all diseased cohorts. Of those three features, PWD was the most discriminative single feature.

Figure 7 compares the influence of different fibrotic LA volume fraction on the mean of the peak-to-peak amplitude change with respect to the healthy baseline case in each of the 12 ECG leads. The amplitudes in aVR, aVL, I, V5 and V6 decreased with an increasing volume fraction of fibrotic tissue in the LA. The maximum reduction was 23% compared to the healthy case. On the other hand, the amplitudes in V3 and III remained nearly unchanged for any fibrotic LA volume fraction.

### 3.3. Estimating the Amount of Fibrosis with Neural Networks

When providing the regression neural network set up as explained in Section 2.5 with all 15 extracted P wave features, the LA fibrotic volume fraction was estimated with an absolute RMSE of 8.78%. Figure 8a shows the predicted volume fraction of fibrosis by the network. The scatter points represents the samples in the test split and their face colors encode their respective total LA volume.

The latter indicates that the network performance decreased for samples with extraordinary high and low LA volumes. A systematic under- and overestimation for low and high LA volumes, respectively, prevailed. Therefore, five additional non-invasive features representing LA and RA volume, torso volume and torso diameter in anterior-posterior direction in the chest and the abdominal region were included for training the network. Due to the consequentially increasing complexity of the network’s input, the size of the hidden layer was increased to 20. In this case, the RMSE between the predicted and ground truth volume fractions covered by fibrosis decreased to 6.39% (Figure 8b).

The regression results from training the network with additional anatomical measures were used to divide the network’s prediction output into four and two different groups for the purpose of discriminating between each of the four Utah classes and between healthy vs. FAM, respectively. The limits for allocating the samples to the different groups as explained in Section 2.5 are visualized in Figure 8. Automatically detecting samples belonging to the FAM group based on the regression results was possible with an accuracy of 99.59%, a sensitivity of 99.78% and a specificity of 86.89%. The area under the ROC curve (AUC) for discriminating between the healthy and FAM group quantified to 0.95. The confusion matrix for distinguishing between the four Utah stages is depicted in Figure 9.

The values in Figure 9 indicates a lower performance for discriminating between the different Utah stages than for healthy vs. FAM. This lower sensitivity and specificity relation also reflects in the results shown in Figure 8: The interquartile ranges between the cohorts characterized by 0 and 5% fibrotic LA volume—resembling the dividing line between the healthy and FAM group—are much more distinct compared to those between Utah stages II and III (20% and 25% fibrotic LA volume) as well as III and IV (35% and 40% fibrotic LA volume).

## 4. Discussion

In this work, we generated 540,000 simulated 12-lead ECGs of healthy subjects and patients with different LA volume fractions covered by fibrosis. Different atrial and thoracic anatomical models derived from statistical shape models as well as varying orientation angles of the atria within the torso are hallmarks of the virtual cohorts. P wave features including duration, PTF V1, peak-to-peak amplitudes and dispersion were calculated for each signal. The influence of anatomical properties on these features’ variances was compared to the variance provoked by fibrotic infiltration of the atrial tissue of various degrees. None of the investigated features showed distinct ranges for the diseased cohort and different healthy anatomical variations. The value ranges of all features occurring from anatomical variations were greater than or equal to the variance in case of fibrotic infiltration of atrial tissue (compare Figure 5). Thereby, the atrial geometry variations caused altered P wave morphologies resulting in varying values for all features, whereas the torso geometry variation mainly affected P wave amplitudes. Nevertheless, all investigated features were characterized by a systematic change in their values for an increasing volume fraction of fibrotic atrial tissue (compare Figure 6). A neural network provided with the aforementioned features of the simulated data succeeded in estimating the volume fraction of atrial fibrosis with an average error of 8.78% fibrotic LA volume. When also including anatomical measures for atria or torso, the RMSE of the regression network decreased to 6.39%. By comparing the overlapping interquartile ranges in Figure 6 to those in Figure 8, it can be inferred that the volume fraction of fibrotic substrate can be estimated more accurately with ECG-based machine learning approaches than by isolatedly considering single P wave features. Therefore, the network indeed seems to be capable of separating the effects of anatomical variability from the influence of fibrotic substrate on the P wave. By employing all investigated P wave features in combination with additional anatomical measures, we achieved an accuracy of 99.59% for discriminating between the healthy control group and the group characterized by a fibrotic LA volume of ≥5%. Sensitivity and specificity resulted to 99.78% and 86.89%. The AUC quantified to 0.95. These classifications were derived directly from the output of the continuous-variable regression network as opposed to training a machine learning classifier targeting discrete classes.

Yoshizawa et al. [4] reported that new-onset AF could be estimated using P wave amplitude in lead II and P wave dispersion features with a sensitivity of 69.1% and specificity of 88.2% in their clinical study comprising 68 AF patients and the same number of controls. Lankveld et al. [16] drew on several time- and frequency domain features to predict AF recurrence rates after PVI in 93 patients with an AUC of 0.76. In the clinical study conducted by Nakatani et al. [15], a combination of several P wave amplitudes led to a sensitivity of 69%, a specificity of 88% and an AUC of 0.77 for predicting the presence of low voltage areas ≥10% in 50 AF patients. Jadidi et al. [5] found that a PWD above 150 ms—provided a very low threshold is set for detecting the P wave offset—was a predictor for advanced LA low voltage substrate with a sensitivity of 94.3% and a specificity of 91.7%. Conte et al. [39] report that PWD and beat-to-beat variability of P wave morphologies held the highest discriminative power to identify patients holding an increased risk of AF occurrence.

Especially sensitivity and AUC results for discriminating between the healthy and the FAM group were higher in our simulation study compared to the findings in previous clinical studies. Possible reasons could involve the number of P waves included in our work (540,000 simulated P waves compared to ≤100 ECG recordings in clinical studies) leading to a larger database for the network to learn relations between P wave features and fibrotic LA volume fraction. Moreover, we found that additional anatomical measures improve the estimation outcome and included them for assessing the regression performance. On the other hand, we investigated in detecting fibrotic substrate in the atria, whereas most clinical studies focused on predicting AF recurrence rates and new-onset AF. Even though the latter are reported to correlate with the amount of the fibrotic LA volume fraction, there is no clear 1:1 relation between them complicating the comparison between our study and those conducted by Yoshizawa et al. [4] and Lankveld et al. [16]. Besides that, the chosen set of investigated P wave features in this study resembles but does not exactly equal the one used in previous clinical studies. We indeed chose the set of features based on previous findings, but used on the one hand a combination of a vast number of features compared to clinical studies [4,16]. On the other hand, we could not evaluate beat-to-beat alterations as proposed by Conte et al. [39] since our simulation setup only allows for a single beat analysis. Furthermore, the impact of noise being superimposed to the actual ECG signal and no CV variations included in the healthy baseline case in our simulations could lead to a set of P waves not capturing the full variability observed in practice and therefore slightly overestimate prediction accuracy.

The electrodes for the lateral ECG leads V5 and V6 are located closest to the LA lateral wall. According to the findings in [26], which we based the distribution of fibrosis in our models on, the presence of fibrotic tissue in this region holds a considerable high probability. Therefore, the amplitude decrease in V5 and V6 that we found for an increasing volume fraction of LA fibrosis can be explained by a growing amount of passive fibrotic elements not contributing to the overall source distribution in the left PV antrum and the LA lateral wall. A careful choice of the threshold for detecting the P wave ending in each lead is crucial as already reported by [5]. In this simulation study, we could choose a simple amplitude threshold of 1.5 × 10${}^{-4}$ mV. However, when changing this threshold value to $3\times 10^{-3}$ mV, PWD does not show the steady increase for different fibrotic LA volume fractions as shown in Figure 6. In this case, the low voltage signal parts at the end of the P wave caused by delayed activation of fibrotic regions in the LA are ignored, which results in underestimation of PWD (see Appendix A). This implies that a sufficiently high signal-to-noise ratio is required for clinically recorded signals in order to apply sensitive thresholds to accurately measure PWD. PTF V1 values were decreasing for an increasing volume fraction of fibrosis covering the total LA volume. Even though the duration of the negative P wave deflection increased in V1, its absolute amplitude simultaneously decreased by an even higher factor. Given that the actual influence of fibrosis on the ECG resembles the changes observed in our simulations, it can be inferred that the volume fraction of atrial fibrosis can be predicted by a neural network with an error of 8.78% on average based on a combination of all examined ECG feature values. Furthermore, we suggest to provide the network with additional anatomical measures non-invasively acquirable by estimating the torso volume with measurements of the torso perimeter and the atrial diameter based, e.g., on echocardiographic recordings to further increase the prediction accuracy.

This study is based on simulated data. Even though we chose an established modeling methodology for fibrotic tissue covering versatile aspects of electrical and structural remodeling on cell and tissue level, different modeling methods could affect the results [29]. Furthermore, the fast iterative method combined with an action potential template as well as the infinite volume conductor method both rely on assumptions. Full bidomain, pseudo-bidomain or reaction-Eikonal [40] simulations could account for diffusion effects neglected when deriving the source distribution by only solving the Eikonal equation and shifting pre-computed action potentials in time. However, the former requires considerable longer computation times compared to the approach used in this study. We therefore intentionally decided for this approach for the sake of reducing computational cost to make the generation of a large database feasible. Moreover, no noise was added to the simulated data analyzed in this study [41]. While noise and a QRS complex immediately following the P wave will definitely impede the feature extraction if averaging over several beats is not feasible, we did not focus on developing robust feature extraction and signal processing methods. We rather wanted to probe the general potential of these features as predictors for the presence of atrial fibrosis provided that their values can be accurately extracted from the ECG. Additionally, we only used one regionally heterogeneous set of CVs while inter-individual CV variation in healthy subjects causes additional P wave feature variability in all model cohorts.

P waves features, especially PWD, resulting from such a variation should be included in the regression model to evaluate whether slower but still physiological CV values yield PWDs in the same value ranges as observed for the diseased cohorts. In this context, one could also examine if the proposed method is capable of detecting fibrotic tissue only in the case of locally heterogeneous conduction velocities or if it is also sensitive towards ionic remodeling and percolation effects and link the results to potentially arrhythmogenic substrate areas. Furthermore, the sensitivity of the neural network’s prediction output to errors in the input feature values is relevant. As shown in Appendix A, determining the PWD is prone to errors since sensitive thresholds are necessary to accurately capture all signal parts belonging to the ECG. Furthermore, measuring the volume of the atria, potentially serving as an additional input feature quantifiable non-invasively via echocardiography, is oftentimes inaccurate. Further future directions could imply to investigate how the specific location of fibrotic patches influences the accuracy of the estimation of left and right atrial fibrosis. Since a large-scale database of >500,000 P waves from healthy and diseased cohorts is now available, it could also be worth to use the signals directly as an input for a convolutional neural network to estimate the fibrotic LA volume fraction without prior feature extraction. In this way, the network could directly learn relations between the different cohorts and derive rules to distinguish between them. Additionally, the effect of different ablation strategies (PVI vs. additional ablation targets) and their effect on the 12-lead ECG could be examined in a future study and reveal important results regarding individual therapy planning. For clinical translation, the trained regression model should be tested prospectively with clinical ECG recordings in the next step to evaluate its performance in comparison with, e.g., LGE-MRI or voltage mapping. Ultimately, the estimated fibrosis fraction could also be predictive for clinical endpoints.

## 5. Conclusions

All investigated P wave features (duration, dispersion, PTF V1 and peak-to-peak amplitudes) were characterized by a large variability in the healthy cohort. For all features, the variances caused by healthy anatomical variation exceeded those resulting from the presence of atrial fibrosis. All P wave features were systematically in- or decreasing for an increasing amount of fibrotic LA volume fraction. However, the value ranges among the different model cohorts overlapped. A neural network provided with a combination of all ECG features and optionally with additional anatomical measures, was successful in estimating the fibrotic LA volume fraction with an absolute RMSE of down to 6.39% Even though clinical translation is challenging and still requires future work regarding developing robust feature extraction methods and drawing conclusions for improving treatment outcomes, the simulation results suggest that the application of our proposed method to clinical data could in future prove valuable for diagnosing fibrotic atrial cardiomyopathy, AF risk stratification and therapy planning.

## Figures and Tables

**Figure 1 jcm-10-01797-f001:**
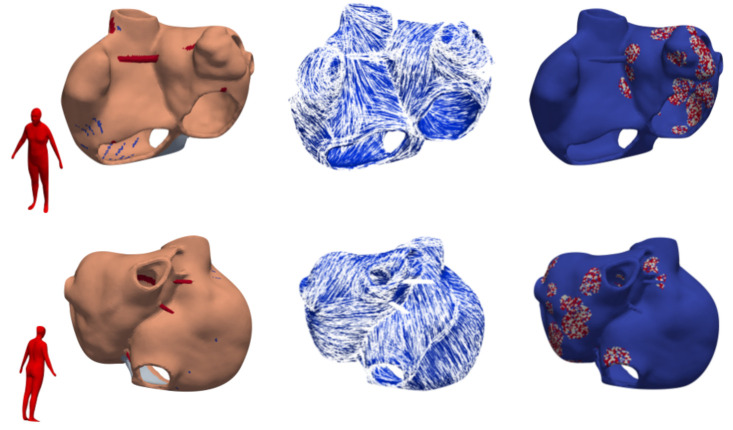
Exemplary volumetric atrial model with labels for different anatomical structures (left column) that define the heterogeneous conduction velocity setup (Section 2.3). Rule-based fiber orientation (middle column) and exemplary fibrosis distribution representing 30% left atrium (LA) fibrotic volume fraction (right column) are depicted aside. The point of view on the atrial models is indicated by the red human body schematics on the left.

**Figure 2 jcm-10-01797-f002:**
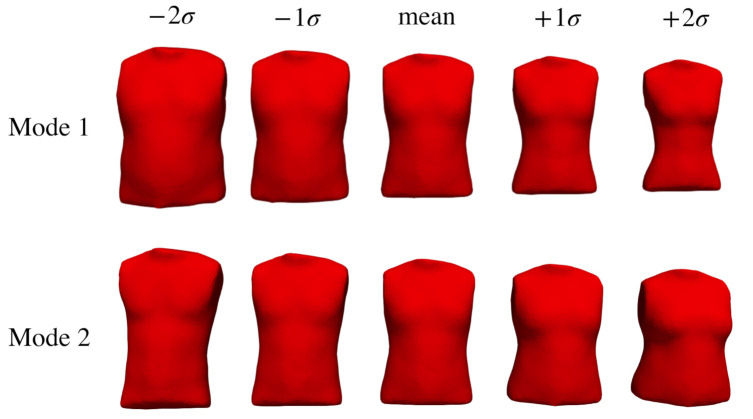
Representation of eigenmodes of the upper body statistical shape model. The first eigenmode (**top** row) reflects in a change of the torso size predominantly in the abdominal region, the second one (**bottom** row) in the chest region.

**Figure 3 jcm-10-01797-f003:**
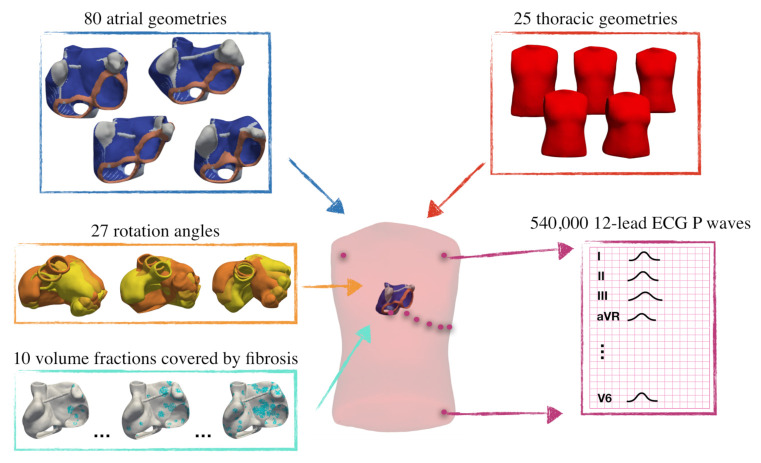
Representation of the different model combinations leading to 540,000 simulated P waves from a virtual patient cohort with fibrosis covering different volume fractions of the atrial tissue.

**Figure 4 jcm-10-01797-f004:**
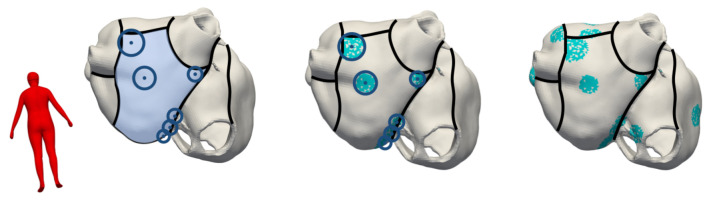
Definition of the spatial distribution of fibrotic tissue. The atrial geometry was first separated into 6 subregions for the left and 2 subregions for the right atrium as reported by Akoum et al. [24] and Higuchi et al. [26] indicated by the black separation lines. The stage of fibrosis to be modeled was then set (15% in this example) and the number of seed points and radii around them were chosen pseudo-randomly by ensuring that within each of these subregions, the total volume of fibrotic elements accumulated to the spatial fibrosis distribution found in clinical studies [24,26]. 80% of the cells in these candidate regions were defined as fibrotic (middle column) and this process was repeated for all subregions (right column).

**Figure 5 jcm-10-01797-f005:**
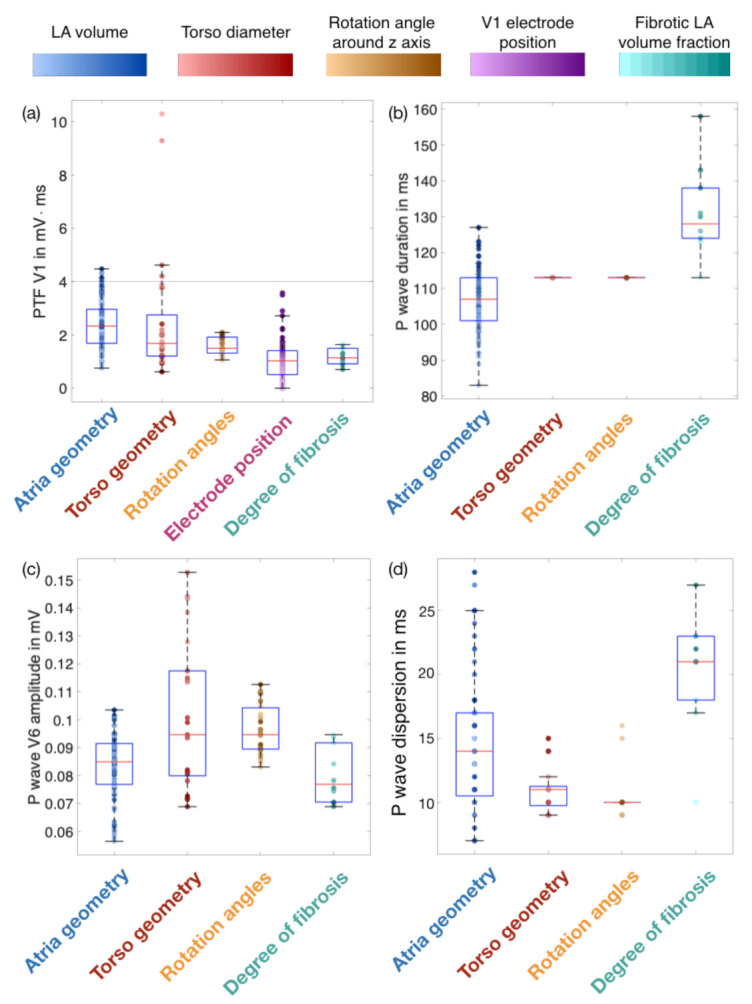
Effect of atrial geometry, thoracic geometry, atrial rotation angles, V1 electrode placement and the fibrotic LA volume fraction on (**a**) P wave terminal force in lead V1 (PTF V1, top left), (**b**) P wave duration (PWD, top right), (**c**) P wave amplitude in V6 (bottom left) and (**d**) P wave dispersion (bottom right). The vertical line in (**a**) indicates the common PTF threshold value of 4 mV·ms. The colored sample points indicate one major change resulting from a variation of the respective influencing factor which consist of the total LA volume (small to large LA volume from light to dark blue), the torso diameter in anterior-posterior direction (small to large diameter from light to dark red), the rotation angle around the z-axis (small to large angle from light to dark orange), the position of the V1 electrode along the inferior-superior direction (inferior to superior direction from light to dark purple) and the fibrotic LA volume fraction (0–45% from light to dark turquoise).

**Figure 6 jcm-10-01797-f006:**
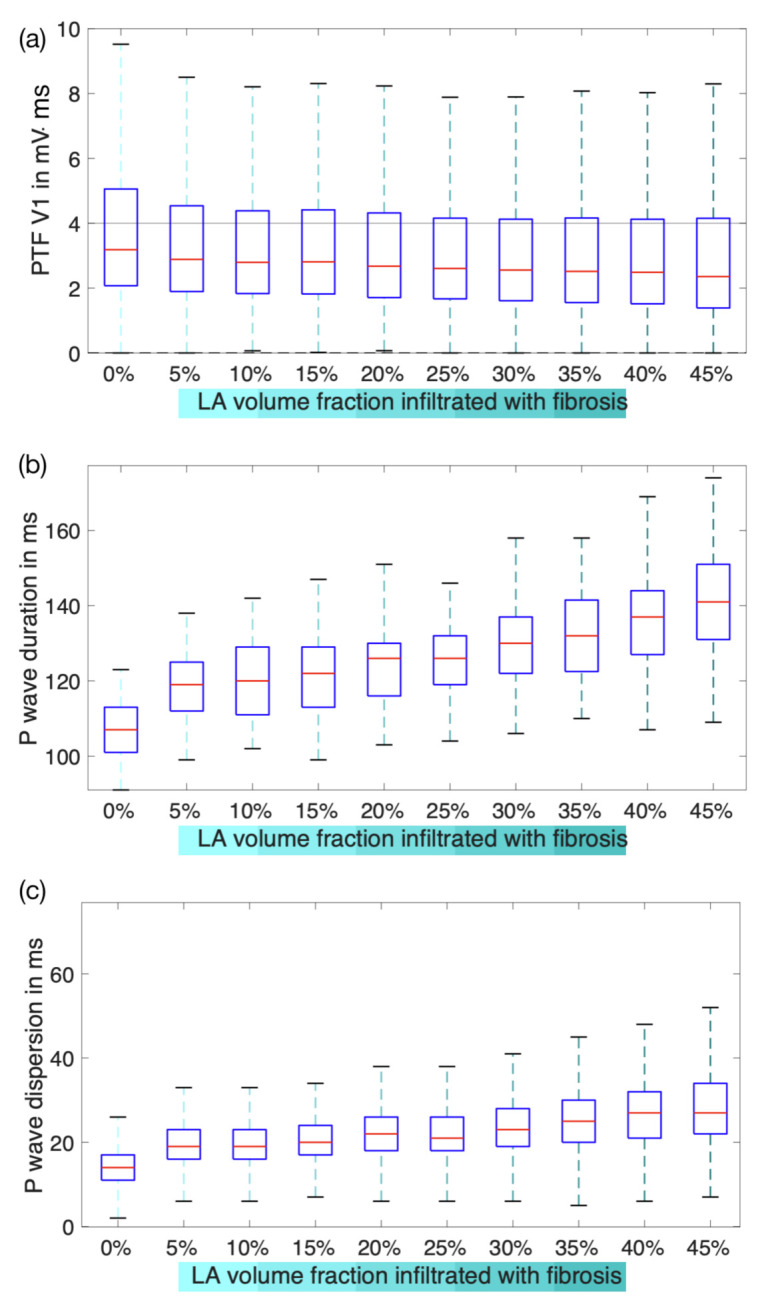
Effect of LA volume fraction infiltrated with fibrosis on (**a**) PTF V1 (top), (**b**) PWD (middle) and (**c**) P wave dispersion (bottom). Each box contains all torso and atrial geometries and rotation angles.

**Figure 7 jcm-10-01797-f007:**
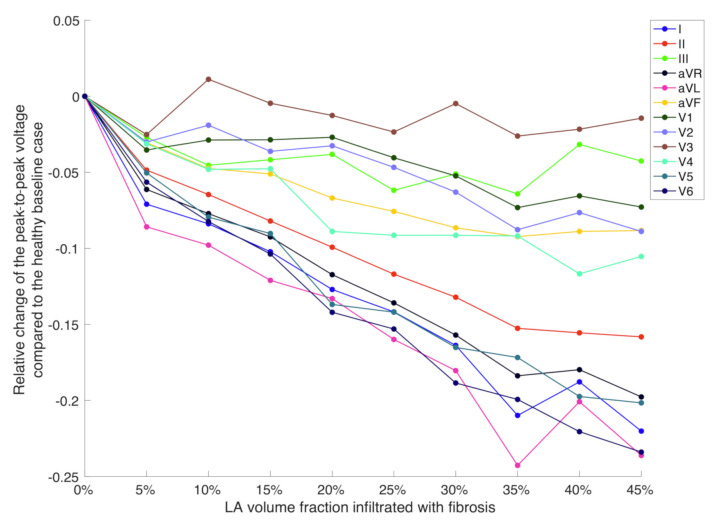
Difference of the peak-to-peak amplitudes in each lead between the healthy baseline case and the mean of each fibrotic model cohort. Each cohort comprised all torso sizes, atrial geometries and rotation angles.

**Figure 8 jcm-10-01797-f008:**
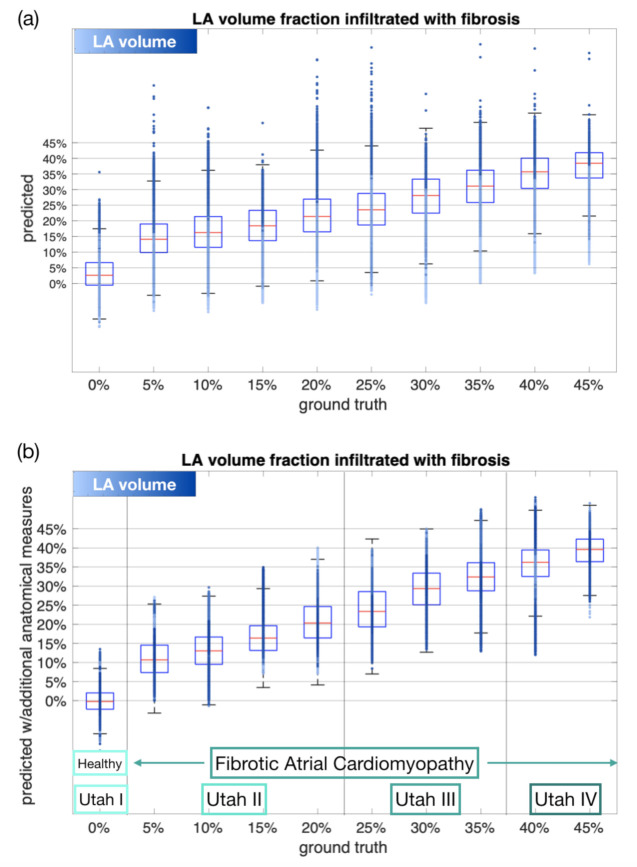
Network performance for predicting the volume fraction of atrial fibrosis (**a**) only based on P wave derived features (top) and (**b**) by also including other non-invasive anatomical measures for atrial and thoracic size (bottom). The color code represents the LA volume of each individual sample point (light: small volume; dark: high volume). The allocation of the different model cohorts into four Utah stages and into healthy vs. fibrotic atrial cardiomyopathy (FAM) is indicated by the vertical lines in the bottom plot.

**Figure 9 jcm-10-01797-f009:**
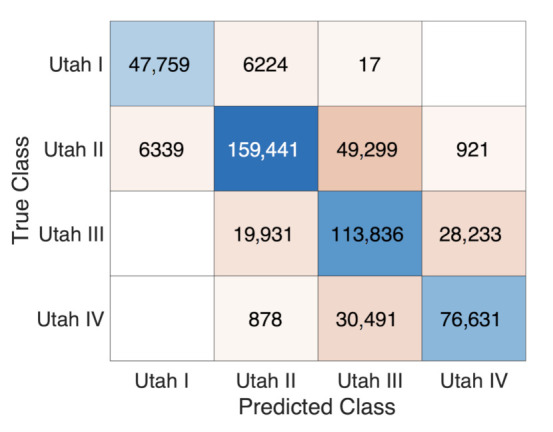
Confusion matrix for discriminating between the four Utah stages.

**Table 1 jcm-10-01797-t001:** Volume fraction of fibrosis in the right atrium as reported by Akoum et al. [24]. LA, left atrium; RA, right atrium.

Utah Stage	Fibrotic LA Volume Fraction	Fibrotic RA Volume Fraction
Utah I	0–5%	1.27 ± 0.38%
Utah II	5–20%	4.65 ± 0.70%
Utah III	20–35%	9.40 ± 2.16%
Utah IV	>35%	12.66 ± 3.0%

**Table 2 jcm-10-01797-t002:** Conduction velocities (CV) in transversal fiber direction and anisotropy ratios.

Tissue Region	CV${}_{⊥}$ in m/s	Anisotropy Ratio (AR)
Bulk right and left atrium	0.591	2.090
Crista terminalis	0.591	2.843
Pectinate muscles	0.461	3.780
Inter-atrial connections	0.645	3.339
Inferior isthmus	0.540	1
Fibrosis (non-conductive)	0	NA
Fibrosis (slow conducting)	0.2 × CV${}_{⊥}$	2.5 × AR

## Data Availability

The statistical shape model of the atria is openly available under creative commons license CC-BY 4.0 together with 100 exemplary volumetric models derived from it including fiber orientation, inter-atrial bridges and material tags [21]. Further data is available from the corresponding author upon reasonable request.

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
