# Peer review of "Non-Invasive and Quantitative Estimation of Left Atrial Fibrosis Based on P Waves of the 12-Lead ECG—A Large-Scale Computational Study Covering Anatomical Variability"

_jcm, 2021, doi:10.3390/jcm10081797_

Round 1

Reviewer 1 Report

Nagel et al. present a comprehensive simulation study varying atrial anatomy, torso geometry, and fibrosis volume fraction for a large number of permutations. They investigated whether fibrosis volume fraction can be inferred from features of the P-wave. The authors should be commended for a large, thorough simulation study and a very nice use of a statistical shape model. I have some suggested changes and comments as follows. 

  • Line 12: change amplitudes to amplitude
  • I personally found the introduction too long
  • Line 67: change nevertheless to however 
  • Figure 1 and section 2.2. It isn't clear how the fibrosis distribution was defined. It would be helpful to include a figure showing how the seed points were defined and the size of the seed areas. Were these seed locations the same between atria, or different? Please make this clear as this is important for interpreting the findings of this study. 
  • How different were fibrosis distributions between cases?
  • Line 186: isoelectric typo 
  • Line 201: for testing rather than for test 
  • How important is the atria vs the torso in the findings? 
  • The train-validation-test split was defined such that an atria can only be in one of sets. I wonder if this should also be the case for the torsos as it looks like torso has the largest effect on results? 
  • Did you try looking at LA fibrosis volume rather than LA fibrosis volume fraction? What is the effect of total atrial volume. 
  • I can see from Figs 1 & 3 that there is also fibrosis in the RA. How did this affect the results? 
  • Figure 4 - it isn't clear how the points are ordered for each parameter. 
  • Lines 283-285 - are these volumes typically available? 
  • Figure 5 - are any of these statistically significant 
  • I think it is outside of this study, but it would be interesting to see how fibrosis location affects the accuracy of the estimation. 
  • Again, outside of this study, it would be very interesting to know if the method can differentiate between types of fibrosis (conductivity changes vs ionic vs percolation/replacement etc). Is the method more able to detect changes in CV or changes in refractory period, or changes in heterogeneity of conduction, and then linking this to potentially arrhythmogenic substrates. Could be a nice future study! 

Reviewer 2 Report

This experimental research used neural network including all ECG features and some disturbing anatomical features to predict the percentage of fibrosis in the atrium and could show that this method leads in these circumstances in satisfactory accuracy.

I have to congratulate the authors for the interesting and well performed investigation. On the other hand, there is still a long way to go to show that even the combinations of ECG features with best analytical methods could lead to clinically significant prediction of clinical outcomes in patients at risk of AF or its complications.

Reviewer 3 Report

Summary:

 The authors analyzed 540,000 simulated 12-lead ECGs and found the changes in P wave features that can be attributed specifically to fibrotic infiltration of atrial tissue. The effects on the P wave could be separated from confounding changes induced by healthy anatomical variation of the atria and the thorax and different electrode positions.

The fibrotic LA volume fraction had a small effect on PTF V1 value, which was more affected by torso geometry. Electrical and structural remodeling of fibrotic tissue resulted in PWDs up to 160ms, particularly in lead V6. PWD was the most discriminative feature between healthy controls and FAM. The amplitudes in aVR, aVL, I, V5 and V6 decreased with an increasing volume fraction of fibrotic tissue in the LA.

A neural network succeeded in estimating the volume fraction of atrial fibrosis. The network seemed to be capable of separating the effects of anatomical variability from the influence of fibrotic substrate on the P wave.

Strength of the work:

- Large simulated cohort with multiple variations randomly chosen reflecting the clinical variability of anatomy and structural remodeling. 

- Fibrosis variation took into account not only the amount of fibrosis but also the distribution, percolation and patchy nature of fibrosis.

- Good methodology and statistical analysis

Comments:  

- Very interesting work showing how simulation models could help predicting prognosis and planning the optimal therapy.

- The evidence for the value and impact of atrial fibrosis as detected by MRI is increasing, however it remains controversial due to technical issues. Another interesting parameter could be the extent of left atrial low voltage area, as detected by atrial mapping and its impact on P wave features.

- As a perspective the effect of different ablation strategies (different lines, fibrosis ablation etc.) and their impact on P waves could be examined.
